# Anti-Inflammatory and Analgesic Effects of TRPV1 Polypeptide Modulator APHC3 in Models of Osteo- and Rheumatoid Arthritis

**DOI:** 10.3390/md19010039

**Published:** 2021-01-17

**Authors:** Yulia A. Logashina, Yulia A. Palikova, Viktor A. Palikov, Vitaly A. Kazakov, Sviatlana V. Smolskaya, Igor A. Dyachenko, Nadezhda V. Tarasova, Yaroslav A. Andreev

**Affiliations:** 1Shemyakin-Ovchinnikov Institute of Bioorganic Chemistry, Russian Academy of Sciences, ul. Miklukho-Maklaya 16/10, 117997 Moscow, Russia; logashina_yu_a@staff.sechenov.ru; 2Institute of Molecular Medicine, Sechenov First Moscow State Medical University, Trubetskaya str. 8, bld. 2, 119991 Moscow, Russia; smolskaya_s_v@staff.sechenov.ru (S.V.S.); tarasova_n_v@staff.sechenov.ru (N.V.T.); 3Branch of the Shemyakin-Ovchinnikov Institute of Bioorganic Chemistry, Russian Academy of Sciences, Prospekt Nauki, 6, 142290 Pushchino, Russia; yuliyapalikova@bibch.ru (Y.A.P.); vpalikov@bibch.ru (V.A.P.); kazakov@bibch.ru (V.A.K.); dyachenko@bibch.ru (I.A.D.)

**Keywords:** arthritis, TRPV1, polypeptide modulator, APHC3, cytokines, inflammation, non-steroidal anti-inflammatory drugs, sea anemone, *Heteractis crispa*

## Abstract

Arthritis is a widespread inflammatory disease associated with progressive articular surface degradation, ongoing pain, and hyperalgesia causing the development of functional limitations and disability. TRPV1 channel is one of the high-potential targets for the treatment of inflammatory diseases. Polypeptide APHC3 from sea anemone *Heteractis crispa* is a mode-selective TRPV1 antagonist that causes mild hypothermia and shows significant anti-inflammatory and analgesic activity in different models of pain. We evaluated the anti-inflammatory properties of APHC3 in models of monosodium iodoacetate (MIA)-induced osteoarthritis and complete Freund’s adjuvant (CFA)-induced rheumatoid monoarthritis in comparison with commonly used non-steroidal anti-inflammatory drugs (NSAIDs) such as diclofenac, ibuprofen, and meloxicam. Subcutaneous administration of APHC3 (0.1 mg/kg) significantly reversed joint swelling, disability, grip strength impairment, and thermal and mechanical hypersensitivity. The effect of APHC3 was equal to or better than that of reference NSAIDs. Protracted treatment with APHC3 decreased IL-1b concentration in synovial fluid, reduced inflammatory changes in joints, and prevented the progression of cartilage degradation. Therefore, polypeptide APHC3 has the potential to be an analgesic and anti-inflammatory substance for the alleviation of arthritis symptoms.

## 1. Introduction

Arthritis is an inflammatory disease characterized by articular surface degradation and ongoing pain and hyperalgesia. The most common types of arthritis are osteoarthritis (OA), which causes chronic pain affecting more than 13% of the adult population, and rheumatoid arthritis (RA), an autoimmune disease affecting more than 1% of the world population. OA is characterized by the progressive degeneration of the articular cartilage, leading to peri-articular bone, synovial joint, and adjusted tissue elements modification [1,2]. The etiology of OA is complex: the initiation and development of OA are triggered by pro-inflammatory cytokines, such as interleukins, in particular IL-1β, and tumor necrosis factor TNF-α, and by generated reactive oxygen species (ROS) [2]. Conventional pharmaceutical treatment of OA patients is focused on pain relief and is usually based on the application of corticosteroids and nonsteroidal anti-inflammatory drugs (NSAIDs), primarily ibuprofen and diclofenac inhibiting both cyclooxygenase isoforms COX-1 and COX-2, and meloxicam, preferentially blocking COX-2 and thus affecting the conversion of arachidonic acid into prostaglandins [3]. RA is a chronic and systemic autoimmune disease affecting multiple symmetric joints. It is characterized by progressive disability and multiple systemic complications, such as cardiovascular, pulmonary, psychological, and skeletal disorders, leading to premature death [4,5]. The etiology of RA is complicated and involves the production of autoantigens, such as rheumatoid factor (RF) and anti-citrullinated protein antibodies (ACPAs), by synovial tissue B-cells during pre-RA stage, and infiltration of synovium by mononuclear cells (activated T- and B-cells) and macrophages releasing cytokines, chemokines, adhesion molecules, matrix metalloproteinases (MMPs), tissue inhibitors of metalloproteinases (TIMPs), and oxygen and nitrogen species during inflammatory stage [6]. Application of disease-modifying anti-rheumatic drugs (DMARDs, e.g., methotrexate [7], hydroxychloroquine [8]), NSAIDs, and steroids leads to multiple negative side effects due to their influence on immune, endocrine, and paracrine systems, thus additional pharmaceutical compounds are applied for arthritis treatment. Among them are drugs targeted to the specific pro-inflammatory cytokines, such as Anakinra, recombinant IL-1 receptor antagonist [9]; ABT-981 (lukitizumab), human anti-IL-1α/β dual immunoglobulin [10]; adalimumab, human anti-TNF-α monoclonal antibody [11,12], and many others [5,13].

Additional pharmaceutical targets have been explored for OA and RA treatments, and one of the promising ones is TRPV1 channel. The role of TRPV1 in nociception and joint inflammation in arthritis has been proven by a number of investigations. Firstly, TRPV1 knocked-out mice have been observed to have attenuated development of adjuvant-induced RA [14,15]. The genetic variant I585V TRPV1 is associated with lower thermal hyperalgesia and an overall risk of symptomatic OA development [16]. Secondly, increased TRPV1 expression in both mRNA and protein levels was observed in synovial fibroblasts [17] and chondrocytes [18] from patients with arthritis, and TRPV1 activation has been shown to upregulate the expression of pro-inflammatory cytokines (IL-1β, IL-6, TNF-α) [16,17] and to be involved in the generation of ROS [19,20]. On the other hand, pro-inflammatory mediators and ROS activate TRPV1, promote transportation and insertion of TRPV1 from the subcellular vesicles pool, and upregulate TRPV1 expression [20,21], indicating that TRPV1 is involved in positive feedback signaling in the process of joint inflammation. Finally, application of TRPV1 agonists and some blockers with pronounced ability to reduce pain and joint inflammation also indicates the important role of TRPV1 in arthritis. The analgesic effect of capsaicin has long been known and used to prevent arthritic pain in animal models [22,23,24]. The attenuation of pain behavior in arthritis animals models has been demonstrated for a variety of TRPV1 antagonists, such as JNJ-17203212 [25], SB366791 [26], and some others.

The first peptide antagonists of TRPV1 with pronounced analgesic activity were isolated from the sea anemone *Heteractis crispa* [27]. Three short polypeptides of 56 amino acids, referred to as analgesic polypeptide, APHC1–3, differing by one (V31P for APHC2) and four (R12P, D23N, V31P, A52G for APHC3) amino acid substitutions were reported [27,28]. Among these polypeptides, APHC3 demonstrated the maximum inhibition level of capsaicin-induced response measured by the patch-clamp technique in whole-cell configuration using CHO cells, which was estimated to be 71 ± 6% at IC_50_ = 18 ± 4 nM, superior to APHC1 and APHC2 with maximal inhibiting levels of 31 ± 9% at IC50 = 60 ± 20 and 42 ± 12% at IC50 = 23 ± 9 nM, respectively [29]. Thorough electrophysiology analysis revealed that APHC polypeptides could either potentiate or inhibit TRPV1 response depending on the strength of the activation stimuli [29]. APHC1–3 have also been demonstrated to reduce high-temperature-induced acute pain using the in vivo hot plate test [30,31]. One of the most remarkable features of APHC polypeptides is their ability to drop the core body temperature [31]. The ability of antagonists to block proton-induced TRPV1 activation is considered to be associated with hyperthermia in vivo [32,33], whereas antagonists that potentiate pH-induced TRPV1 activity have a hypothermic or no effect on the core body temperature [34,35]. APHC3 polypeptide has been shown to either reversibly inhibit acid-induced Ca^2+^ influx or to potentiate TRPV1 response to acidic pH depending on the experimental conditions [29,31]. APHC3 application at doses 0.1 and 0.5 mg/kg had a moderate hypothermic effect with a body temperature decrease of 0.6 °C and 0.4 °C, respectively, while homologous polypeptide APHC1 at the same doses produced a significant decrease in body temperature of 0.8 °C and 2.1 °C [31]. The analgesic effect of APHC1 and APHC3 polypeptides at 0.1–0.5 mg/kg doses has been confirmed in vivo in acute pain (hot plate, capsaicin-induced pain test, acetic acid-induced writhing) and chronic pain models (formalin, CFA-induced hyperalgesia) [31].

Considering the ability of APHC3 polypeptide to modulate pH-induced TRPV1 response and its strong analgesic effect on the inflammatory phase of the formalin-induced pain model, we suggested that this antagonist can be successfully applied for arthritic pain relief. The ability of APHC3 to suppress ankle joint inflammation and to inhibit thermal and mechanical hyperalgesia, associated with arthritis, was elevated by the usage of two rat models of arthritis: complete Freund’s adjuvant (CFA)-induced RA and monosodium iodoacetate (MIA)-induced OA [36,37]. The joint destruction during OA has been shown to depend on the level of proinflammatory cytokine IL-1β in synovial fluid [38]. To elucidate the effect of APHC3 on IL-1β levels we performed an immunoassay of synovial fluid from MIA-induced OA rats.

## 2. Results

### 2.1. CFA-Induced Monoarthritis

#### 2.1.1. Assessment of Inflammation In Vivo

The degree of ankle joint inflammation in vivo was evaluated by joint swelling and local temperature. CFA injection caused an increase of joint diameter by 2–3 mm on day 3 in groups treated with saline, APHC3 in doses, 0.01 and 0.05 mg/kg, diclofenac, and ibuprofen as compared to control. Joint diameter in groups treated with APHC3 at doses of 0.1 and 1 mg/kg did not differ from the control group (Figure 1a and Appendix A).

Ratios of the treated to intact joint diameters were 20–25% higher in groups treated with saline, APHC3 0.01 mg/kg, and diclofenac than in the control group. In groups treated with ibuprofen or APHC3 0.05–1 mg/kg, joint diameter ratios were not changed as compared to control (Figure 1a). The joint local temperature was similar among the experimental groups (Appendix A).

#### 2.1.2. Assessment of Locomotor Activity

We tested possible effects of APHC3 treatment on locomotor activity due to its anti-inflammatory action in the CFA arthritis rat model. Neither arthritis induction with CFA nor treatment with APHC3 or comparison drugs changed parameters of horizontal and vertical locomotor activity in an open field in any of the groups (Appendix A).

#### 2.1.3. Behavioral Assessment of Pain Sensitivity

In vivo assessment of hypersensitivity to thermal and mechanical stimuli and pain-induced functional disability after CFA arthritis induction allowed us to estimate the model’s effectiveness. Besides this, behavioral testing is a valuable tool for the evaluation of the anti-inflammatory effects of APHC3 treatment.

CFA significantly reduced hot plate paw withdrawal latency measured on day 3 after injection. However, it did not differ between the control animals in the groups receiving APHC3 and diclofenac or ibuprofen. Treatment with APHC3 0.05 and 0.1 mg/kg almost doubled paw withdrawal latency as compared to the group treated with saline after CFA injection (Figure 1b). Therefore, APHC3 effectively reversed thermal nociceptive hypersensitivity, which is consistent with our previous data [31].

Mechanical hyperalgesia was measured as a paw withdrawal response to a gradual increase of mechanical pressure applied by the pincher analgesia meter. In contrast to the control group, the pain threshold of compression was reduced more than 2-fold in the saline, APHC3 0.01 mg/kg, and in groups treated with both comparison drugs. In the groups treated with 0.05–1 mg/kg APHC3, we did not find mechanical hyperalgesia ((Figure 1c). Moreover, APHC3 in doses 0.1 and 1 mg/kg significantly increased the paw withdrawal threshold as compared to the group that received saline after CFA injection (Figure 1c).

Significant hindlimb grip strength deficiency developed in groups treated with saline and diclofenac after CFA arthritis induction. APHC3 and ibuprofen successfully reversed pain-induced paw dysfunction. Grip strength in the saline-injected group was significantly lower than in groups administered with 0.1 and 1 mg/kg APHC3 (Figure 1d).

### 2.2. MIA-Induced Arthritis

Assessments of inflammation in vivo and functional tests were conducted on days 3, 7, and 14 after OA induction with MIA injection into the right knee joint. Thermal and mechanical hypersensitivity and pain-induced articular disability were tested in this model. For evaluation of APHC3 anti-inflammatory effect on MIA-induced arthritis, 0.01 and 0.1 mg/kg doses were selected.

#### 2.2.1. Assessment of Inflammation In Vivo

Inflammation was assessed in vivo as the increase of injected joint diameter and the ratio of injected to intact joint diameters (in percent of the intact joint). A significant increase of the injected joint diameter due to swelling was registered only on day 3 in groups administered with saline, APHC3 0.01 mg/kg, ibuprofen, and meloxicam (Appendix A). The diameter of the intact joint did not change during the experiment (Appendix A). The ratio of injected to intact joint diameter increased most prominently also on day 3 after OA induction with a progressive decline in time (Figure 2a–c). The ratio increased by 10–15% in comparison to control on day 3 in groups administered with saline, APHC3 0.01 mg/kg, and both comparison drugs (Figure 2a), whereas only on day 7 in the group administered with saline after OA induction was it larger than in control (Figure 2b). On day 14 no significant differences were detected among tested groups (Figure 2c). At the same time, administration of 0.1 mg/kg APHC3 markedly reduced joint inflammation even on day 3 after MIA injection compared to the saline-treated group (Figure 2a).

#### 2.2.2. The Concentration of Proinflammatory Cytokine IL-1β

The primary role of IL-1β in OA development is based on its ability to enhance the synthesis of proteolytic molecules involved in cartilage degradation and upregulate other inflammatory mediators expression, including TNF-α, IL-8, prostaglandin E2 PGE2, COX-2, MMPs, and other [39,40].

Levels of IL-1β in synovial fluid were significantly higher in the group treated with saline after OA induction than in the control group both on days 8 and 15. Treatment with meloxicam prevented IL-1β rise on day 8, however, on day 15 its level increased significantly compared to control. On the contrary, in a group administered with 0.1 mg/kg APHC3, IL-1β concentration was significantly higher than in control on day 8 and did not differ from it on day 15 (Figure 3).

#### 2.2.3. Assessment of Pain-Related Behavior

Mechanical allodynia was tested by the von Frey method which detects pain threshold as the paw withdrawal response. Rats that received saline after OA induction showed a marked decrease of hind paw withdrawal threshold, denoting the presence of tactile allodynia. This decrease was maintained throughout all 14 days of the study proving the effectiveness of the protocol used. On the other hand, treatment with both doses of APHC3 or comparison drugs meloxicam and ibuprofen effectively reversed mechanical allodynia even after the first administration on day 3. Paw withdrawal thresholds in these groups did not differ from the control group and were significantly higher than in the saline-treated group. This dynamic was maintained until day 14 of the study (Figure 4a–c).

Assessment of thermal hyperalgesia was conducted with a hot plate test on day 14 after OA induction. In contrast to mechanical sensitivity, there were no significant differences revealed in thermal sensitivity among the studied groups at this time point (Figure 4d).

In addition to sensitivity changes caused by OA induction, we tested articular dysfunction related to pain sensation.

Pain-induced articular discomfort was evaluated in the incapacitance tester which measures the weight-bearing differences between arthritic and intact hind limbs. In the control group, animal weight was distributed equally between two paws. MIA injection followed by saline treatment resulted in a significant decrease of injected limb loading on days 3 and 7 compared to the control group (Figure 5a,b). The most prominent difference between intact and injected limbs, about 45%, in a group treated with saline was registered on day 7 after OA induction (Figure 5b). It is worth noting that 0.1 mg/kg APHC3 and ibuprofen effectively reversed pain-induced knee joint incapacitation after the first administration on day 3, while 0.01 mg/kg APHC3 and meloxicam did not. Weight-bearing in the group treated with 0.01 mg/kg APHC3 did not differ from the control group on day 7 (Figure 5b). Finally, on day 14 of testing, there were no signs of weight-bearing deficits identified among the studied groups (Figure 5c).

Functional disability estimated in grip strength test on days 3 and 7 demonstrated results similar to the incapacitation test. In particular, significant grip strength deficits were shown in groups treated with saline and meloxicam with the approximate levels constituting 50 and 70% of the control group, respectively. At the same time, grip strength in groups treated with APHC3 in both tested doses and ibuprofen did not differ from the control group but were higher than in the saline-treated group during the whole testing period (Figure 6).

#### 2.2.4. Knee Joint Histology

For the histological analysis, injected knee joints were harvested on days 8 and 15 after MIA arthritis induction. Characteristic histological signs of OA, such as synovial inflammation and hyperplasia, as well as destruction of the bone tissue and articular cartilage, were analyzed.

MIA injection into the knee joint led to progressive destruction of the normal joint tissues, indicating the effectiveness of the used OA model. Indeed, inflammatory infiltration and hyperplasia of synovia and destruction of both the distal femoral and proximal tibial cartilage were revealed in all OA groups. The most noticeable arthritic histological signs were observed in the group treated with saline in comparison to the control group. APHC3 0.1 mg/kg, meloxicam, and ibuprofen showed comparable effectiveness in the prevention of inflammation-induced histological changes of the synovia, while APHC3 0.01 mg/kg was ineffective (Figure 7). At the same time, APHC3 0.1 mg/kg appears to be more efficient than both comparison drugs and APHC3 0.01 mg/kg for suppression of articular cartilage destruction (Figure 8). Representative images illustrating synovitis and synovial hyperplasia for each group on day 15 are shown in Figure 9 and Appendix A.

It is noteworthy that there were no signs of bone tissue destruction identified on day 8. On day 15, minor destructive changes were observed under the periosteum. They were associated with surrounding soft tissue inflammation, but not with joint cartilage destruction (Figure 9, Appendix A). All tested compounds reduced bone destructive changes, but the period of observation after OA induction was too short for adequate evaluation (Appendix A).

## 3. Discussion

As the major nocisensor, TRPV1 is considered an attractive pharmacological target for the development of analgesic and anti-inflammatory compounds [41]. TRPV1-selective agonists, such as capsaicin, produce transient channel activation and Ca^2+^ influx followed by desensitization with analgesic effects [42,43]. However, the clinical application of TRPV1 agonists is limited because of the pain and the neurotoxic side effects correlated with the channel activity [44,45].

TRPV1-selective antagonists could overcome the negative side effects due to their ability to block channel activity. Although the usage of TRPV1-selective antagonists as a pain killer is considered to be beneficial, none of them have yet been approved for the clinical trial third phase either due to severe side effects [46,47] or due to the absence of noticeable efficacy (AZD1386, NEO6860) (https://clinicaltrials.gov/). Nonetheless, the search for appropriate TRPV1 antagonists continues.

TRPV1 antagonists are considered to be two types: polymodal TRPV1 antagonists, which hinder all activation modes of TRPV1, and mode-selective ones, which efficiently block activation by capsaicin, but can produce variable effects (including either potentiation, no effect, or low-potency inhibition) by the proton and/or heat activation modes [35]. Polymodal TRPV1 antagonists have been tested in models of arthritis with controversial results. Intra-articular (1 mg) and systemic (~6 mg/kg, i.p.) administration of JNJ-17203212 reduced pain behaviors in the MIA-induced model of arthritis pain. Systemic administration of AMG9810 (30 mg/kg, i.p.) reversed thermal hyperalgesia and partially reversed MIA-induced change in weight-bearing after a 3 mg dose of MIA, but did not affect a conditioned place preference assay [48]. No reduction in guarding behavior in a CFA-induced arthritis model was found after the oral administration of the TRPV1 antagonist AZD1386 [49].

None of the mode-selective TRPV1 antagonists have been tested in models of CFA- and MIA-induced arthritis. In the present work, we investigated the analgesic and anti-inflammatory effects of APHC3, a polypeptide modulator of TRPV1 channel, in two rat models of arthritis. Previously the action of APHC3 on TRPV1 in vitro was found to depend both on the nature of the activation stimuli and on the strength of the stimuli. APHC3 mainly potentiated TRPV1 response to low activation strength stimuli of capsaicin, while at increasing activation strength the potentiating effects disappeared or switched to inhibition [29]. It is noteworthy that APHC3 inhibited the response of TRPV1 to combined stimuli pH + capsaicin, which is more relevant to activation stimuli in the site of inflammation. Arthritis is characterized by hypersensitivity to thermal and mechanical stimuli accompanied by pain-induced functional disability [50,51]. We found that APHC3 significantly alleviated inflammation-associated arthritic symptoms, such as joint swelling, pain-induced behavior, and hypersensitivity to the various stimuli in rats with CFA- or MIA-induced arthritis.

CFA injected into the joint provokes an inflammatory response mediated by the immune system, mimicking rheumatoid arthritis (RA) when immune cells such as T cells and B cells and macrophages infiltrate the joints and cause pain, swelling, and stiffness [49]. CFA activates the innate immune system but is not adaptive, therefore this model may not provide the optimal conditions for delineation of the mechanisms of RA development [52]. Collagen-induced arthritis is one of the most disease-related and widely used models of RA [52]. CFA-induced monoarthritis may be efficiently used for the analysis of novel anti-inflammatory and analgesic drugs suitable for arthritis symptomatic treatment [49]. Intra-articular injection of CFA leads to infiltration of inflammatory cells and synovial hypertrophy and is generally accepted as a RA model. However, it is important to note that it significantly differs from the histological point of view because common aspects of human RA such as bone erosion and cartilage serration are usually absent [52]. In the CFA-induced monoarthritis model, the pain and inflammation severity reaches the maximum on day 1 and 2 after CFA injection and on day 3 begins to reduce [51]. This simulates the early stage of RA in humans that often starts from acute inflammation of one joint. Therefore we started treatment at the peak of symptom severity and analyzed the effects at the end of the maximum severity interval. In our experiments, CFA injection into the knee joint caused significant swelling of the joint, thermal and mechanical hyperalgesia, and reduced hindlimb strength (Figure 1) as was previously described [49,51]. We did not find significant changes in joint temperature and parameters of locomotor activity in the open field test after CFA injection (Appendix A). NSAIDs are used to alleviate pain and inflammation in RA treatment, therefore we tested commonly used drugs, diclofenac and ibuprofen, at doses close to the maximum recommended in humans, as a positive control. Subcutaneous injection of APHC3 for 3 days dose-dependently reversed the inflammatory effects of CFA injection. The effect reached a maximum at dose 0.1 mg/kg while the dose 1 mg/kg did not provide additional benefits. At 0.1 and 1 mg/kg APHC3 significantly reversed joint swelling (~80% compared to saline-treated group) when both non-selective COX inhibitors (ibuprofen and diclofenac) were ineffective (Figure 1a). TRPV1 is greatly involved in thermal hypersensitivity generated by inflammation [53] including arthritis-induced thermal hypersensitivity [48]. APHC3 efficiently reversed thermal hypersensitivity in CFA-induced arthritis at doses higher than 0.05 mg/kg while diclofenac was almost ineffective, and ibuprofen showed moderate efficacy (Figure 1b). We observed the same distribution of efficacy in the hindlimb grip strength test, highlighting the link between hypersensitivity and the ability to use the limb. (Figure 1b,d). Both ibuprofen and diclofenac were unable to reverse mechanical hypersensitivity following CFA injection. APHC3 dose-dependently reversed mechanical hypersensitivity (Figure 1c) confirming the significant role of TRPV1 activation in this process, as was shown previously on TRPV1 knockout mice [14].

OA is one of the most common joint diseases, characterized by degeneration of articular cartilage, subchondral bone sclerosis, secondary synovitis, and chronic joint pain, which significantly decrease patients’ quality of life. Tissue inflammation accompanied by pain and molecular and structural alterations of the extracellular matrix, which reduces joint flexibility, are the hallmarks of OA [54,55].

The MIA-induced OA model is considered to reproduce OA processes in humans [56,57]. MIA injection into the rat knee joint provokes inflammation and degenerative changes (cartilage degradation, subchondral bone changes, synovial inflammation) [58,59]. Pain behaviors in the animal model are easily acquired (weight-bearing pain, tactile allodynia, and mechanical hyperalgesia) and reflect movement-induced pain in patients with OA [60].

We compared APHC3, a mode selective antagonist of TRPV1, with ibuprofen (non-selective COX-1 and 2 inhibitor) and meloxicam (a selective COX-2 inhibitor). NSAIDs are still the most commonly recommended and used drugs in OA treatment, despite being often insufficient to relieve pain [61]. The doses of APHC3 were chosen as the most effective (0.1 mg/kg, s.c.) and minimally effective (0.01 mg/kg, s.c.) according to efficacy in CFA-induced arthritis and previous results [31]. The doses of ibuprofen and meloxicam were chosen as relevant to the maximum recommended doses for patient treatment [61,62].

On day 3 after MIA injection, joint inflammation and pain-related behavior were assessed 60 min after first-time compound/saline administration, which reflects a single dose effect. APHC3 at 0.1 mg/kg almost completely reversed joint inflammation, supporting the important role of TRPV1 and neurogenic inflammation in this process (Figure 2a). Neither meloxicam nor ibuprofen were able to reduce inflammation after single-dose administration. All tested compounds completely reversed mechanical hypersensitivity after the first administration and during the time of the experiment (Figure 4). Both doses of APHC3 and ibuprofen, but not meloxicam, significantly reversed disability and improved grip strength after single-dose administration on day 3 (Figure 5a and Figure 6a). Therefore, a single dose of APHC3 and ibuprofen produced a significant analgesic effect to reverse disability and weakening of grip strength associated with movement-induced pain, whereas the selective COX-2 inhibitor meloxicam was able only to reduce mechanical hypersensitivity in the von Frey test.

The effects of regular administration of compounds were assessed on day 7 (5 days of the treatment) and day 14 (12 days of the treatment). We found significant effects of MIA injection on mechanical hypersensitivity and grip strength impairment at all time points of the experiment, but joint diameters and functional disability were significantly different only on days 3 and 7. All tested compounds reduced joint diameter on day 7, confirming their anti-inflammatory properties (Figure 2b). Regular administration of APHC3 and ibuprofen effectively reversed functional disability and the grip strength impairment, whereas meloxicam was almost ineffective (Figure 5 and Figure 6). Therefore, the effects of single-dose and regular treatment were very similar at the level of pain tests, but regular treatment by COX inhibitors reduced inflammation of joints better than a single dose.

Selective COX-2 inhibitors show better safety, tolerability, and efficacy for the treatment of OA than non-selective COX inhibitors both in animals [49,63] and humans [61,62]. The reason for their efficacy is a more pronounced activity in the site of inflammation. Meloxicam showed much better efficacy than ibuprofen in the reduction of joint inflammation from a histological point of view (Figure 7, Figure 8 and Figure 9), as was reported previously [63], and therefore, it could have more benefits in the treatment of chronic disease. APHC3 (0.1 mg/kg) had similar effects as COX inhibitors on synovitis, medium efficacy on hyperplasia (better than ibuprofen, but worse than meloxicam) (Figure 7), but it was the only compound in our study that prevented progression of articular cartilage destruction (Figure 8c,d).

Pro-inflammatory molecules (cytokines or neuropeptides) accumulate in affected joints during OA and interact with their receptors on sensory neurons, sensitizing and activating TRPV1-mediating nociceptive and neuropathic pain responses. Most probably, APHC3, acting as an inhibitor of highly intensive TRPV1 activation, reversed thermal hypersensitivity and neurogenic inflammation by reducing the release of proinflammatory peptides (CGRP, substance P, etc.) from peptidergic sensory neurons in the injured joint. Additionally, we found that APHC3-induced decrease of neurogenic inflammation led to a reduction of IL-1b concentration in synovial fluid after regular administration for 12 days (Figure 3). The role of IL-1b is highly discussed in the development and progression of osteoarthritis. It induces the expression of MMPs responsible for cartilage degradation in OA and suppresses proteoglycan synthesis [64]. In surgical models of OA, IL-1b is unambiguously involved in the development of the disease, especially in the early stages [64]. In our experiments, we found an increase of IL-1b after induction of OA, but the controversy of previously reported results for this model should be noted [65,66]. The diverse action of meloxicam and APHC3 on IL-1b concentration in synovial fluid is very intriguing. Meloxicam produced a significant decrease of IL-1b concentrations at the early stage of OA development (day 8) and did not affect them on day 15. While APHC3 significantly decreased IL-1b concentration on day 15, it was ineffective on day 8. Therefore, APHC3 prevents propagation of the inflammatory response, decreasing the level of major interleukin responsible for cartilage degeneration. Histology analysis revealed that APHC3 efficiently prevented inflammatory changes in the joint and protected cartilage from degradation. At 0.1 mg/kg, APHC3 suppressed the articular cartilage destruction that occurred in all other groups from day 8 to day 15.

Meloxicam produced similar or better remission of joint inflammation than AHPC3 (except cartilage destruction) but had significantly worse efficacy in the reversal of disability and the impairment of grip strength. Analysis of clinical usage of COX inhibitors for the treatment of OA-related pain showed that only 50% of patients can expect substantial pain relief [61]. Multimodal TRPV1 antagonists have controversial efficacy, but TRPV1 agonists are proven to be effective in the treatment of OA-related pain [67,68]. The major difference of APHC3 from multimodal TRPV1 antagonists is the ability to potentiate responses to acidic pH and low strength stimuli [29]. This dependence on activation stimuli and activation strength manifests itself in a moderate hypothermic effect in vivo, but can also explain the robust analgesic and anti-inflammatory activities described for this polypeptide [28,31]. Studies of functionally related channel TRPA1 [69] revealed that weak activators of this channel [70,71,72] and potentiators [73,74] can promote the defunctionalization of TRPA1-expressing neurons by reducing voltage-gated calcium and sodium currents. The weak activation of TRPA1 was considered a promising strategy to alleviate pain. Therefore, we can suggest that APHC3 can affect TRPV1-expressing neurons subjected to weak activation stimuli outside of affected joints, and decrease their excitability in the same manner as described for TRPV1 agonists [75]. Additional depolarization block of sensory neurons can be useful to prevent or reduce the development of the neuropathic component that plays a significant role in OA-related pain [76,77].

## 4. Materials and Methods

### 4.1. Ethics Statement

This study conforms fully to the World Health Organization’s International Guiding Principles for Biomedical Research Involving Animals. All experiments were approved by the Institutional Commission for the Control and Use of Laboratory Animals of the Branch of the Shemyakin-Ovchinnikov Institute of Bioorganic Chemistry of the Russian Academy of Sciences (protocol number: 688/19, date of approval: 17 January 2019).

### 4.2. Drugs

APHC3 was produced as described previously [78]. Diclofenac sodium salt, ibuprofen, and meloxicam were purchased from Sigma-Aldrich (Moscow, Russia).

### 4.3. Animals

Experiments were performed on 8–10-week-old male Sprague Dawley rats (Animal Breeding Facility Branch of Shemyakin-Ovchinnikov Institute of Bioorganic Chemistry, Russian Academy of Sciences, Pushchino, Russia) weighing 250–270 g. Animals were housed at room temperature (23 ± 2 °C) in a 12 h light–dark cycle with ad libitum access to food and water.

### 4.4. CFA-Induced Monoarthritis Model and Compounds Administration

On day 0 rats were anesthetized with an intramuscular injection of Zoletil (20–40 mg/kg, Virbac Sante Animale) and Xylazine (5–10 mg/kg, Pharmamagist, Ltd., Budapest, Hungary). Freund’s complete adjuvant (CFA, 40 μL, Sigma-Aldrich) was injected intra-articularly into the right ankle joint with the left joint kept intact. The control group (CTRL) received intra-articular saline (40 μL) injection.

Drug administration was carried out daily on days 1–3 after CFA injection (Figure 10a). Saline or APHC3 (0.01, 0.05, 0.1 or 1 mg/kg) were injected subcutaneously (2 mL/kg), ibuprofen (40 mg/kg) was gavaged (10 mL/kg), and diclofenac (20 mg/kg) was injected intramuscularly (1 mL/kg).

Pain-related behavior and joint inflammation were assessed 60 min after compound administration on day 3.

### 4.5. MIA-Induced OA Model and Compound Administration

To induce osteoarthritis with monoiodoacetate (MIA), on day 0 rats were anesthetized with Zoletil and Xylazine as described above for CFA injection. Animals of all groups except for the control group received an intra-articular injection of 3 mg MIA in 50 μL of sterile saline to the right knee joint. The control group was injected with the same volume of sterile saline. The left joint was kept intact both in MIA-injected and CTRL groups.

Test compounds were administered daily from day 3 to day 14 after MIA injection (Figure 10b). Saline or APHC3 (0.01 or 0.1 mg/kg) were injected subcutaneously (2 mL/kg), meloxicam (0.5 mg/kg) was injected intramuscularly (0.3 mL/kg), and ibuprofen (40 mg/kg) was gavaged (10 mL/kg).

Joint inflammation and pain-related behavior were assessed 60 min after compound administration on days 3, 7, and 14.

On days 8 and 15 rats were sacrificed and samples of synovial fluid were collected from the MIA-treated knee joints or joints were dissected for histological analysis.

### 4.6. Assessment of Inflammation In Vivo

Ankle joint diameters were measured in CFA-induced arthritis, and knee joint diameters were measured for MIA-induced arthritis. Joint diameters of both legs were measured using a digital caliper to evaluate swelling degree. Both absolute increase in joint diameter and the ratio between treated and intact joints (in percent of the intact joint) of the same animal were assessed. Joint diameter ratio was calculated according to the following equation: (diameter of injected joint/diameter of intact joint) × 100.

For CFA-induced arthritis, the local temperature of the ankle joint was measured with a non-contact infrared thermometer.

### 4.7. Measurement of IL-1β Concentration in Synovial Fluid

Measurements were conducted in groups treated with saline, APHC3 0.1 mg/kg, and meloxicam after OA induction and in the control group. On days 8 and 15 after knee joint injection, rats were sacrificed, and samples of the synovial fluid were collected via lavage with 100 µL of phosphate-buffered saline with 4 mM EDTA from the injected knee joints and stored at −80 °C until use. The concentration of IL-1β in the synovial fluid was detected using IL-1β Rat ELISA Kit (Invitrogen, Thermo Fisher Scientific, Waltham, MA, USA) according to the manufacturer’s protocol.

### 4.8. Assessment of Locomotor Activity

Spontaneous locomotor activity was recorded for 3 min using a system that counts interruptions in a set of photo beams (OPTO—VARIMEX (Columbus Instruments, Columbus, OH, USA) and ATM3 Auto System using Auto-Track Version 4.2 software (Columbus Instruments, Columbus, OH, USA)). Parameters of the locomotor activity including travelled distance, travelling, and resting time (horizontal activity) and the number of rears (vertical activity) were analyzed.

### 4.9. Assessment of Pain-Related Behavior

#### 4.9.1. Hot Plate Test

Rats subjected to model arthritis can develop hypersensitivity to heat [48,79]. Sensitivity to a thermal stimulus was tested using a Hot-Plate Analgesia Meter (Columbus Instruments, Columbus, OH, USA) set at 55 °C. Rats were placed individually on the preheated hot-plate surface and exposed to heat until nociceptive reaction was registered. Test was discontinued if withdrawal response was not observed for 30 s. Pain threshold was detected as latency to hind paw withdrawal or licking.

#### 4.9.2. Pincher-Based Algometer Test

Mechanical nociceptive stimulation was provided by a pincher-based algometer (BIO-RP-1, Bioseb, Vitrolles, France). Rats were gently restrained with a towel on the bench. Algometer forceps were placed around the hind paw, and increasing pressure was applied. Paw withdrawal response to the applied pressure was registered as a pain threshold to assess mechanical hyperalgesia. Data were averaged from three trials with 1 min inter-application intervals.

#### 4.9.3. Hindlimb Grip Strength Test

Behavioral assessment of movement-evoked pain was carried out in a bilateral hindlimb grip strength test. A Grip Strength Meter (Columbus Instruments, Columbus, OH, USA) consisting of a wire mesh frame connected to the strain gauge was used. Rats were gently restrained and allowed to grasp the wire mesh frame with their hind paws and then were pulled backwards until the grip released. Three measurements were conducted with an interval of 30 s for the averaged grip strength calculation.

#### 4.9.4. Test with Von Frey Filaments

Cutaneous mechanical sensitivity was determined with electronic von Frey filament (BIO-EVF, Bioseb, Vitrolles, France). Before testing, rats were placed for a 10 min adaptation period into plastic chambers with an elevated mesh-bottomed platform allowing access to the plantar surface of the hindpaws. A filament was applied to the plantar surface of the injected hindpaw with gradually increasing pressure until a paw withdrawal response was elicited. Data were averaged from three trials with inter-application intervals of at least 1 min to avoid sensitization to the mechanical stimuli.

#### 4.9.5. Incapacitation Test

The level of discomfort in the arthritic limb was estimated from the weight distribution between rear paws. Rats were positioned in the chamber of the incapacitance tester (BIO-SWB-TOUCH, Bioseb, Vitrolles, France) and allowed to adapt for at least 5 min. Then the force exerted by each hindlimb was evaluated. Final weight distribution was calculated from three trials conducted with intervals of at least 1 min. Data for the arthritic limb were normalized to the intact limb. Therefore, equal weight distribution is indicated by values at about 100%, whereas on the arthritic side it was <100%.

### 4.10. Joint Histology

Rats were sacrificed on days 8 and 15 and treated knee joints were dissected and fixed with 10% buffered formalin for 7 days. Subsequently, joints were decalcified in Trilon B for 7 days, sectioned in the sagittal plane, dehydrated with alcohol, and embedded in paraffin blocks. Then, 5-μm-thick tissue sections were stained with hematoxylin and eosin for analysis of histopathological signs including inflammatory infiltration of synovia (synovitis) and synovial hyperplasia, cartilage damage, and bone resorption. Each sign was graded with scores on a scale of 0 to 5, where 0 represents normal tissue and 5 represents severe tissue degeneration [80]. Images were acquired with an AxioScope.A1 light microscope equipped with an Axiocam 305 color camera and ZEN 2.6 lite software (Carl Zeiss Microscopy GmbH, Oberkochen, Germany).

### 4.11. Statistical Analysis

Statistical analysis was performed using GraphPad Prism 6.0 for Windows (GraphPad Software, San Diego, CA, USA). The non-parametric Kruskal–Wallis test with Dunn’s multiple comparisons post-test was used as appropriate for multiple independent samples. A value of *p* < 0.05 was considered indicative of a statistically significant difference.

On the box plots, data are presented as median, mean (shown with +), first to third interquartile range, minimum, and maximum. Histograms show mean with standard deviation (SD).

## 5. Conclusions

APHC3, the polypeptide modulator of TRPV1 channel, produced significant anti-inflammatory and analgesic effects in models of rheumatoid arthritis and osteoarthritis. The efficacy of APHC3 was higher or equal to commonly used NSAIDs in standard tests on pain-related behavior. APHC3 at dose 0.1 mg/kg showed quick anti-oedemic action and a significant reversal of disability and mechanical hypersensitivity, accompanied by recovery of hindlimb lability and functionality. Additionally, we found that a 12-day treatment with APHC3 decreased IL-1b concentration in synovial fluid. Histological analysis revealed regression of joint inflammation and cartilage degradation compared to non-treated rats. Therefore, we can conclude that polypeptide APHC3 has the potential to be an analgesic and anti-inflammatory substance for the alleviation of arthritis symptoms.

## Figures and Tables

**Figure 1 marinedrugs-19-00039-f001:**
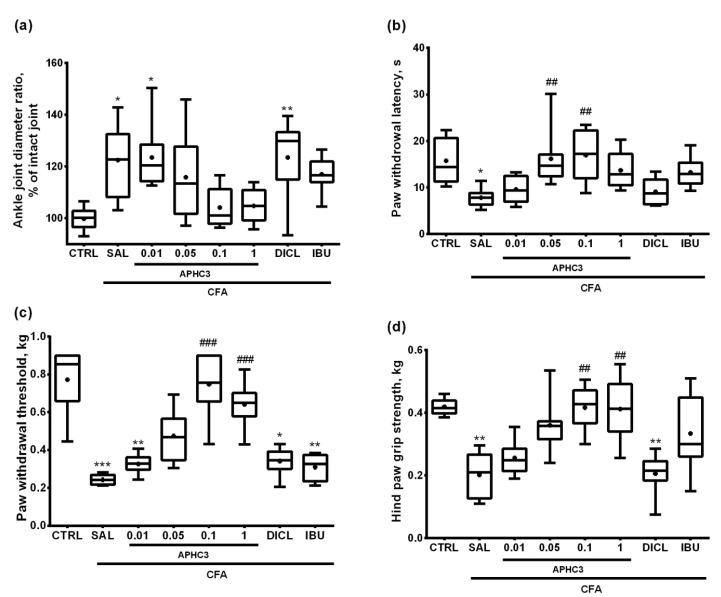
Assessment of inflammation in the ankle joint and pain-related behavior on the day 3 after intra-articular administration of CFA (40 μL) and anti-inflammatory effects of APHC3 (0.01, 0.05, 0.1, and 1 mg/kg s.c.), diclofenac (20 mg/kg i.m.) and ibuprofen (40 mg/kg p.o.). (**a**) Normalized diameters of CFA-injected joints. (**b**) Thermal sensitivity in the hot plate test, (**c**) mechanical nociception in the pincher-based algometer test. (**d**) Movement-evoked pain sensitivity in the hindlimb grip strength test. CTRL, ontrol group, SAL—sterile saline, DICL—diclofenac, and IBU—ibuprofen. Results are presented as median, mean is shown as a cross (+), interquartile range, and minimum and maximum (*n* = 8 for each group). Statistical analysis was performed using the Kruskal–Wallis test followed by Dunn’s multiple comparisons test. *—*p* < 0.05 vs. CTRL, **—*p* < 0.01 vs. CTRL, ***—*p* < 0.001 vs. CTRL, ##—*p* < 0.01 vs. SAL, ###—*p* < 0.001 vs. SAL.

**Figure 2 marinedrugs-19-00039-f002:**
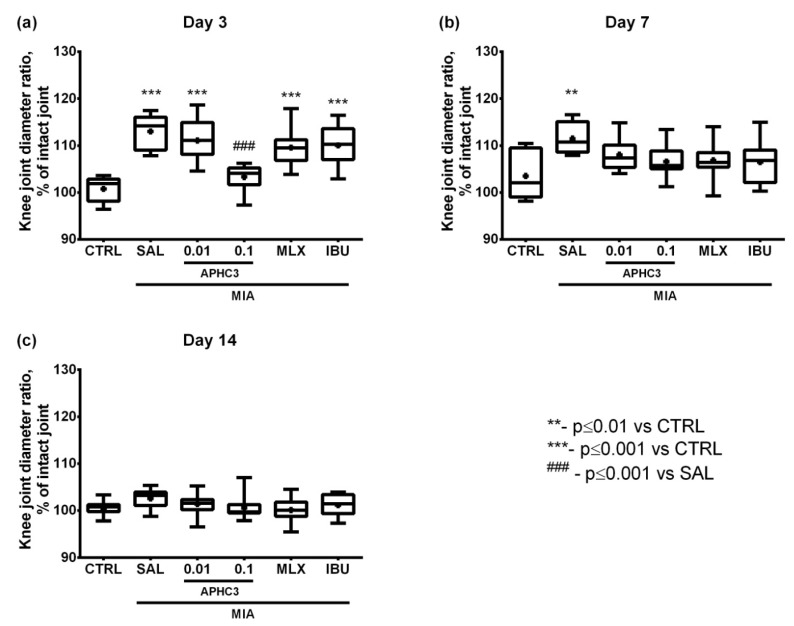
Normalized diameters of the knee joints in the MIA-induced OA model. Diameters (in mm) of the injected and intact (Appendix A) joints were measured on days 3, 7, and 14 after intra-articular MIA injection into the right knee joint (3 mg MIA in 50 μL of sterile saline). Normalized diameters of the injected joint on days 3 (**a**), 7 (**b**), and 14 (**c**) are expressed in percent of intact joint diameters. APHC3 (0.01 and 0.1 mg/kg s.c.), meloxicam (0.5 mg/kg i.m.), and ibuprofen (40 mg/kg p.o.) were administered daily on days 3–14. CTRL—control group, SAL—sterile saline, MLX—meloxicam, and IBU—ibuprofen. Results are presented as median, mean shown as a cross (+), interquartile range, and minimum and maximum (*n* = 10–12 for each group). Statistical analysis was performed using the Kruskal–Wallis test followed by Dunn’s multiple comparisons test. **—*p* < 0.01 vs. CTRL, ***—*p* < 0.001 vs. CTRL, ###—*p* < 0.001 vs. SAL.

**Figure 3 marinedrugs-19-00039-f003:**
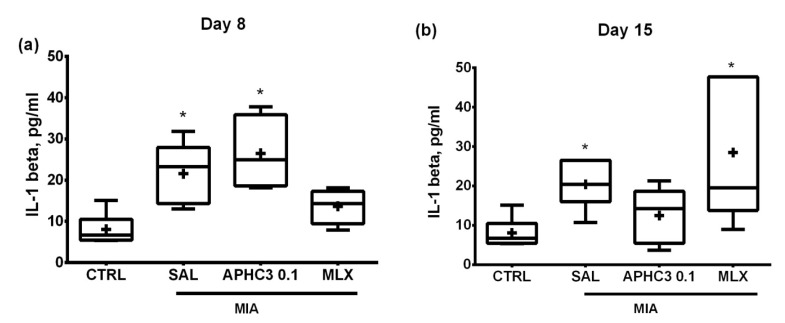
IL-1b concentration (in pg/mL) in the synovial fluid of the injected knee joints in the MIA-induced OA model. Control group (CTRL) and groups administered daily on days 3–14 with saline (SAL), APHC3 0.1 mg/kg s.c., or meloxicam (MLX) 0.5 mg/kg i.m. were chosen for the measurement. Lavage was collected from the synovial cavity on days 8 (**a**) and 15 (**b**) and analyzed for IL-1 beta concentration with ELISA kit. Results are presented as median, mean shown as a cross (+), interquartile range, minimum, and maximum (*n* = 6 for each group). Statistical analysis was performed using the Kruskal–Wallis test followed by Dunn’s multiple comparisons test. *—*p* < 0.05 vs. CTRL.

**Figure 4 marinedrugs-19-00039-f004:**
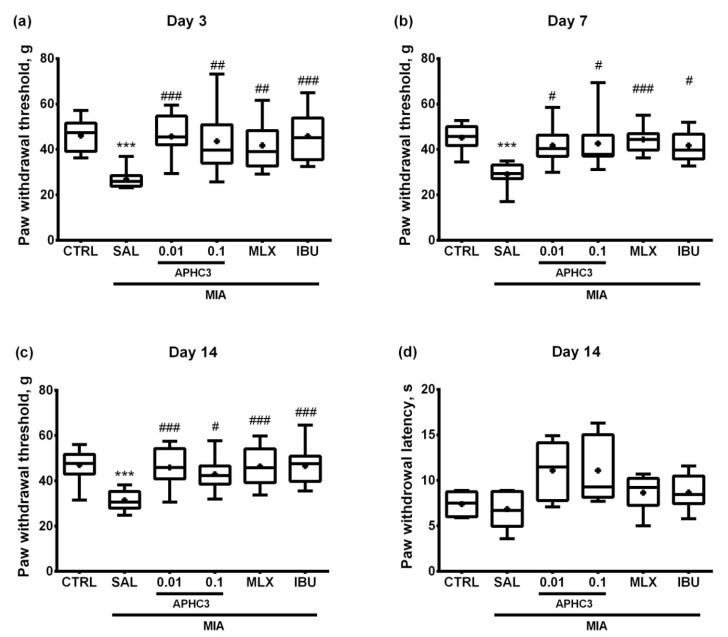
Mechanical allodynia and thermal sensitivity in the MIA-induced OA model. Mechanical allodynia was assessed with electronic von Frey filament on days 3 (**a**), 7 (**b**), and 14 (**c**) after intra-articular MIA injection into the right knee joint (3 mg MIA in 50 μL of sterile saline). Thermal sensitivity (**d**) in the hot plate test was assessed on day 14. APHC3 (0.01 and 0.1 mg/kg s.c.), meloxicam (MLX, 0.5 mg/kg i.m.), and ibuprofen (IBU, 40 mg/kg p.o.) were administered daily on days 3–14. CTRL and SAL designate control and saline-treated groups, respectively. Results are presented as median, mean shown as a cross (+), interquartile range, minimum, and maximum (*n* = 10–12 for each group). Statistical analysis was performed using the Kruskal–Wallis test followed by Dunn’s multiple comparisons test. ***—*p* < 0.001 vs. CTRL, #—*p* < 0.05 vs. SAL, ##—*p* < 0.01 vs. SAL, ###—*p* < 0.001 vs. SAL.

**Figure 5 marinedrugs-19-00039-f005:**
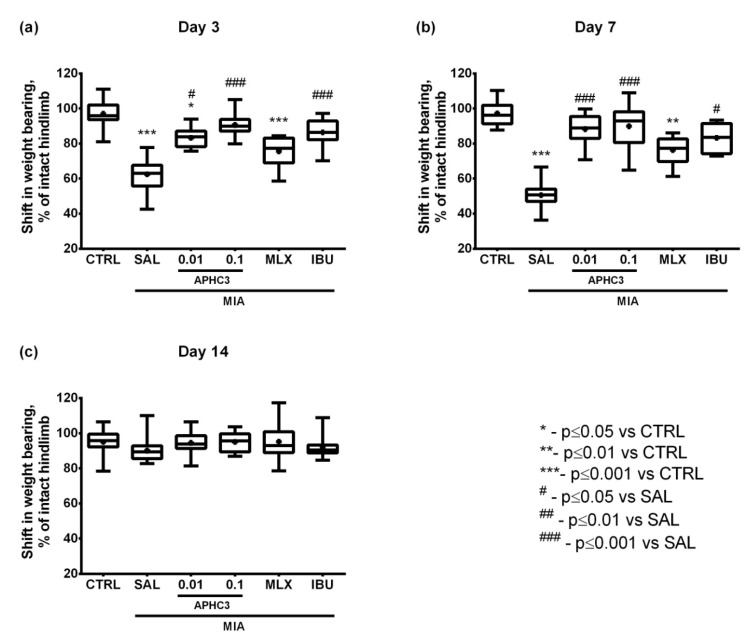
The normalized level of discomfort in the arthritic limb in the MIA-induced OA model. Weight distribution between rear paws was estimated with the incapacitance tester on days 3 (**a**), 7 (**b**), and 14 (**c**) after intra-articular MIA injection into the right knee joint (3 mg MIA in 50 μL of sterile saline). APHC3 (0.01 and 0.1 mg/kg s.c.), meloxicam (MLX, 0.5 mg/kg i.m.), and ibuprofen (IBU, 40 mg/kg p.o.) were administered daily on days 3–14. Abbreviations CTRL and SAL designate control and saline-treated groups, respectively. Results are presented as median, mean shown as a cross (+), interquartile range, minimum, and maximum (*n* = 10–12 for each group). Statistical analysis was performed using the Kruskal–Wallis test followed by Dunn’s multiple comparisons test. *—*p* < 0.05 vs. CTRL, **—*p* < 0.01 vs. CTRL, ***—*p* < 0.001 vs. CTRL, #—*p* < 0.05 vs. SAL, ###—*p* < 0.001 vs. SAL.

**Figure 6 marinedrugs-19-00039-f006:**
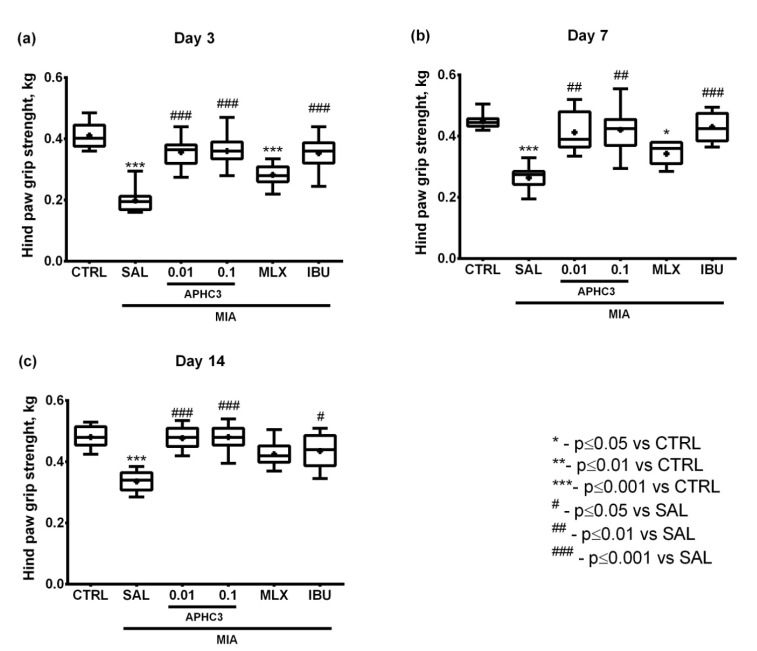
Grip strength of the arthritic limb in the MIA-induced OA model. Grip strength was assessed with a Grip Strength Meter on days 3 (**a**), 7 (**b**), and 14 (**c**) after intra-articular MIA injection into the right knee joint (3 mg MIA in 50 μL of sterile saline). APHC3 (0.01 and 0.1 mg/kg s.c.), meloxicam (MLX, 0.5 mg/kg i.m.), and ibuprofen (IBU, 40 mg/kg p.o.) were administered daily on days 3–14. Control and saline-administered groups are denoted as CTRL and SAL, respectively. Results are presented as median, mean shown as a cross (+), interquartile range, minimum, and maximum (*n* = 10–12 for each group). Statistical analysis was performed using the Kruskal–Wallis test followed by Dunn’s multiple comparisons test. *—*p* < 0.05 vs. CTRL, ***—*p* < 0.001 vs. CTRL, #—*p* < 0.05 vs. SAL, ##—*p* < 0.01 vs. SAL, ###—*p* < 0.001 vs. SAL.

**Figure 7 marinedrugs-19-00039-f007:**
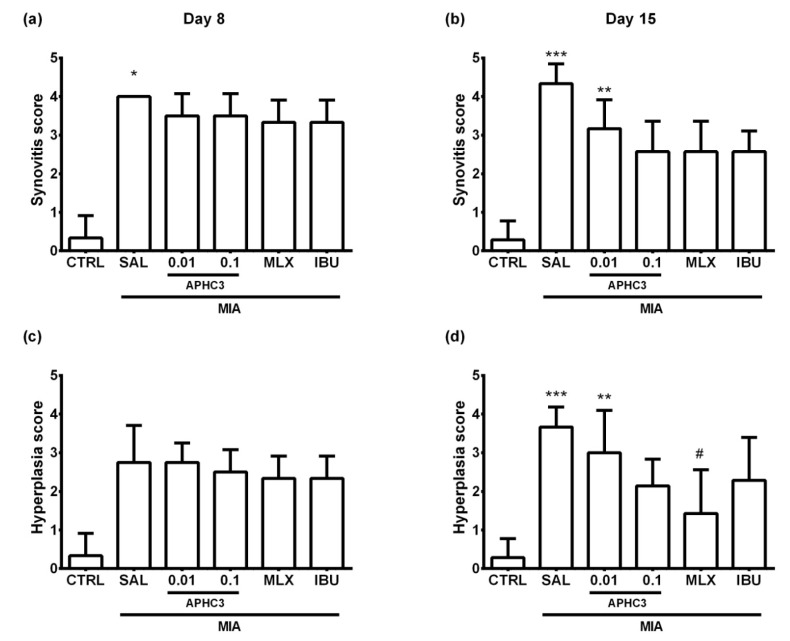
Synovitis and synovial hyperplasia of the injected knee joint in the MIA-induced OA model. Synovitis (**a**,**b**) and synovial hyperplasia (**c**,**d**) were assessed on days 8 (**a**,**c**) and 15 (**b**,**d**) after intra-articular MIA injection into the right knee joint (3 mg MIA in 50 μL of sterile saline). APHC3 (0.01 and 0.1 mg/kg s.c.), meloxicam (MLX, 0.5 mg/kg i.m.), and ibuprofen (IBU, 40 mg/kg p.o.) were administered daily on days 3–14. Abbreviations CTRL and SAL designate control and saline-treated groups, respectively. Results are presented as mean and SD (*n* = 4 for day 8, *n* = 6–7 for day 15). Statistical analysis was performed using the Kruskal–Wallis test followed by Dunn’s multiple comparisons test. *—*p* < 0.05 vs. CTRL, **—*p* < 0.01 vs. CTRL, ***—*p* < 0.001 vs. CTRL, #—*p* < 0.05 vs. SAL.

**Figure 8 marinedrugs-19-00039-f008:**
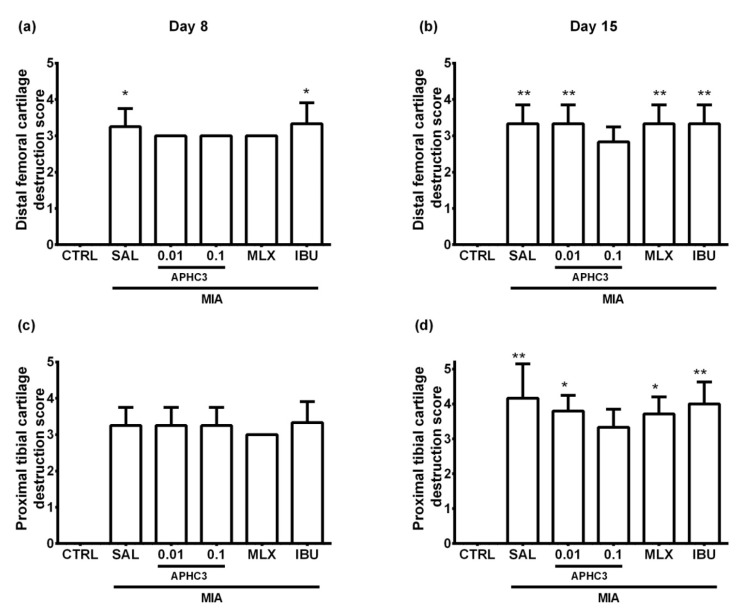
Histological analysis of cartilage destruction of the injected knee joint in the MIA-induced OA model. Destructive changes of the distal femoral (**a**,**b**) and proximal tibial (**c**,**d**) cartilage were assessed on days 8 (**a**,**c**) and 15 (**b**,**d**) after intra-articular MIA injection into the right knee joint (3 mg MIA in 50 μL of sterile saline). APHC3 (0.01 and 0.1 mg/kg s.c.), meloxicam (MLX, 0.5 mg/kg i.m.), and ibuprofen (IBU, 40 mg/kg p.o.) were administered daily on days 3–14. Control and saline-administered groups are denoted as CTRL and SAL, respectively. Results are presented as mean and SD (*n* = 4 for day 8, *n* = 6–7 for day 15). Statistical analysis was performed using the Kruskal–Wallis test followed by Dunn’s multiple comparisons test. *—*p* < 0.05 vs. CTRL, **—*p* < 0.01 vs. CTRL.

**Figure 9 marinedrugs-19-00039-f009:**
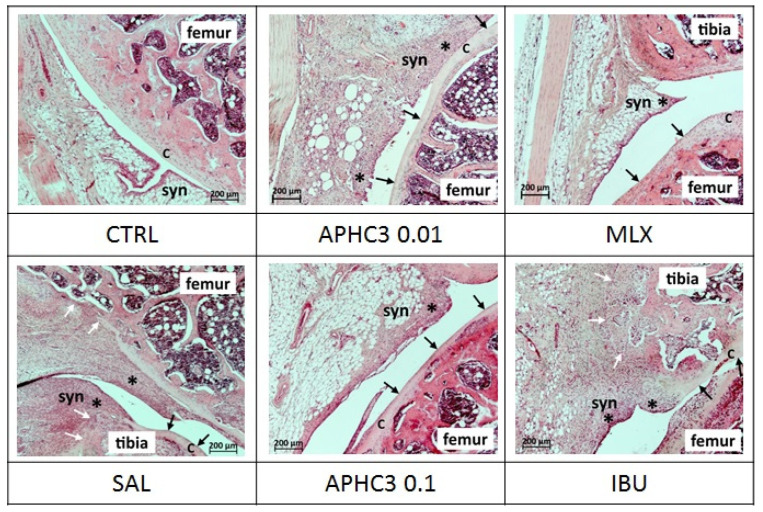
Representative images of the injected knee joint synovia in the MIA-induced OA model (scale bar = 200 µm, 50× magnification). Syn, synovial tissue; C, cartilage; the femur and the tibia are indicated accordingly; *, inflammatory infiltration of synovia; black arrows, destructive changes of cartilage; white arrows, destructive changes of bones. Inflammatory infiltration (InIn) and synovial hyperplasia (SH) were assessed on day 15 after intra-articular MIA injection into the right knee joint (3 mg MIA in 50 μL of sterile saline). Each sign was graded with scores on a scale of 0 to 5, where 0 represents normal tissue and 5 represents severe tissue degeneration. CTRL (control group): InIn 0, SH 0; SAL (sterile saline-treated group): InIn 5, SH 4; APHC3 0.01 mg/kg: InIn 3, SH 3; APHC3 0.1 mg/kg: InIn 2, SH 1; MLX (meloxicam-treated group): InIn 2, SH 0; IBU (ibuprofen-treated group): InIn 3, SH 2.

**Figure 10 marinedrugs-19-00039-f010:**
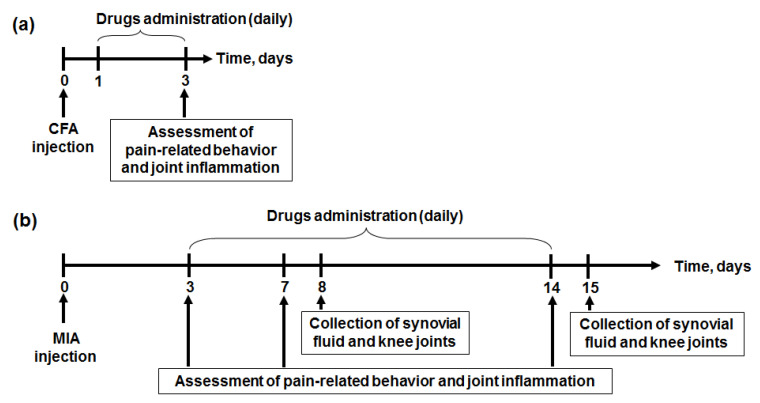
Experimental timeline for CFA-induced (**a**) and MIA-induced (**b**) arthritis models.

## Data Availability

Data is contained within the article or Appendix A.

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
