# Peer review of "Anti-Inflammatory and Analgesic Effects of TRPV1 Polypeptide Modulator APHC3 in Models of Osteo- and Rheumatoid Arthritis"

_marinedrugs, 2021, doi:10.3390/md19010039_

Round 1

Reviewer 1 Report

The manuscript “Anti-inflammatory and analgesic effects of TRPV1polypeptide modulator APHC3 in models of osteo- and rheumatoid arthritis” by Logashina et al is an interesting paper describing the effects of a polypeptide from sea anemone Heteractis crispa, APHC3, in two animal models of Rheumatoid arthritis and Osteoarthritis, respectively. The Authors report that APHC3 is able to counteract the inflammation and the disease progression in both animal models.

Few topics need to be addressed:

1) The number of animals used to perform the experiments needs to be reported.

2) The conclusion should be less emphasized.

3)The English language should be improved throughout the text.

4) The sentence on lanes 50-51 should be rephrased: “RA is a chronic and systemic autoimmune disease affecting multiple symmetric joints including synovium, articular cartilage, and bone” . Synovium, cartilage and bone are not joints but they constitute the joints.

Minor

On line 59 there are too much parentheses

Author Response

Thank you very much for positively considering our submission

Point-by-point discussion

The manuscript “Anti-inflammatory and analgesic effects of TRPV1polypeptide modulator APHC3 in models of osteo- and rheumatoid arthritis” by Logashina et al is an interesting paper describing the effects of a polypeptide from sea anemone Heteractis crispa, APHC3, in two animal models of Rheumatoid arthritis and Osteoarthritis, respectively. The Authors report that APHC3 is able to counteract the inflammation and the disease progression in both animal models.

Few topics need to be addressed:

1) The number of animals used to perform the experiments needs to be reported.

Added to figure legends in the manuscript.

2) The conclusion should be less emphasized.

Some changes were made to conclusions to make it less emphasized. Please see the Conclusion.

3)The English language should be improved throughout the text.

The text of the manuscript was corrected by the proofreading service

4) The sentence on lanes 50-51 should be rephrased: “RA is a chronic and systemic autoimmune disease affecting multiple symmetric joints including synovium, articular cartilage, and bone” . Synovium, cartilage and bone are not joints but they constitute the joints.

Corrected to "RA is a chronic and systemic autoimmune disease affecting multiple symmetric joints"

Minor 

On line 59 there are too much parentheses

Corrected accordingly

Reviewer 2 Report

Authors present in this manuscript the anti-inflammatory and analgesic effect of a TRPV1 antagonist in two models, one for RA and other for OA.

From my point of view it is better to focus in just one of the two model, to be able to focus better in the effect of this molecule on the onset of the disease. In this manuscript, authors focus on day 3 after compounds administration for RA and days 8 and 15 for OA. These time points do not allow to explore the role of the drug in the long course of both diseases. Focus on one disease will allow authors to be able to explose early and long time points what is better to understand the real effect of the drug in the disease. For example, to study better if there are changes in bone erosions that they do not see at this time points, etc.

Moreover, n number of animlas used is missing.

Lines 519-531 must be removed as they are from ´authors guidelines´.

Author Response

Authors present in this manuscript the anti-inflammatory and analgesic effect of a TRPV1 antagonist in two models, one for RA and other for OA.

From my point of view it is better to focus in just one of the two model, to be able to focus better in the effect of this molecule on the onset of the disease.

We admitted that the idea of focusing on one model of disease is reasonable. However, initially we have to perform highly complicated researches to prove the APHC3 effectiveness for the alleviation of arthritis symptoms. Hence, both models of arthritis were investigated. We also intended to analyze the linkage between TRPV1, neurogenic inflammation, and production of cytokines. We would like to note, that based on the obtained results for MIA-induced OA, this model looks very prominent for a more detailed analysis of APHC3 treatment realized in the future.

We compared APHC3 with NSAIDs because these drugs widely but not always successfully used to reduce pain and inflammation in patients with OA and RA. Therefore, the time points were chosen to model the aggravation of disease when symptomatic treatment is essential. Additionally, we described some positive effects of APHC3 and NSAIDs on joints to confirm the applicability of the peptide for the treatment of arthritis.  The CFA-induced RA model of monoarthritis is a relatively quick way to get information about the efficacy of the molecule for the alleviation of arthritis’ symptoms. Additionally, this model gives us information about the most probable minimal and maximal effective doses of APHC3 in arthritis models.  

In this manuscript, authors focus on day 3 after compounds administration for RA and days 8 and 15 for OA. These time points do not allow to explore the role of the drug in the long course of both diseases. Focus on one disease will allow authors to be able to explose early and long time points what is better to understand the real effect of the drug in the disease. For example, to study better if there are changes in bone erosions that they do not see at this time points, etc.

In this manuscript, we evaluated the potential of APHC3 to alleviate the symptoms of arthritis.  APHC3 and NSAIDs caused a significant decrease of inflammation that evidently reduced the progression of the disease. CFA – induced monoarthritis is the thoroughly studied and stable model that is used for validation of the efficacy of novel drugs (https://doi.org/10.1016/j.ejphar.2015.02.050) and new methods verification for arthritis study (https://doi.org/10.1016/j.jneumeth.2017.04.011). It mimics the early stage of RA in humans that often starts from acute inflammation of one joint. In this model, day 3 is the time point when acute inflammation begins to change to chronic and suits well for the assessment of inflammation and pain-related behavior (the standard pain-related behavior is described in https://doi.org/10.1016/j.jneumeth.2017.04.011 fig 7,8,10). We added some information about CFA-induced monoarthritis in the manuscript-  lines 587-598

Moreover, n number of animlas used is missing.

 Added to figures legends

Lines 519-531 must be removed as they are from ´authors guidelines´.

Corrected. We beg a pardon for our carelessness.

Reviewer 3 Report

In this study the authors explore the role of a TRPV1 inhibitor APHC3 in two models of arthritis (extrapolating to both RA and OA), exploring its impact on pain and inflammation. Aspects of the study are performed to a high standard such as measurements of pain. Other elements are limited or insufficient. Particularly for the CFA model. This is used as an rheumatoid arthritis (RA) model and is not up to scratch. Further supporting data is needed to interpret their results in this model including histology, serum/synovial measurements of inflammation. It is not really clear whether it is being used as a model of polyarthritis monoarthritis, and its validity as a model of RA is not clear in this context. I would suggest removing this entirely from the manuscript and focussing on the more robustly assessed MIA model. In addition, the authors need to combine their figs into fewer figs (11 is excessive for this paper), moving repetitive non-significant control data into supplementary would achieve this. The histology in particularly needs to be presented more clearly and comprehensively to support the findings. Lastly the discussion and final conclusions should reflect the merit of APHC3, but be more honest in the appraisal, as the evidence that it significantly outperforms standard NSAIDS here is almost entirely absent. There is some great data here, but it needs to be tidied up and made more concise.

Major

Introduction

Authors state “It is obvious that additional pharmaceutical targets should be explored for the OA and RA treatments, and one of the most prominent is TRPV1 channel”. I think that TRPV1 is an interesting target but in the field of rheumatology I wouldn’t say that it is the most prominent target. Perhaps reword for balance to potential or promising.

In 2006 Barton et al explored the value of inhibition in a monoarthritis model, with some positive results, but this does not appear to have been extensively followed up in further models of arthritis

2.1.1. Assessment of inflammation in vivo. Further clarity in reporting of the different doses is required. “APHC3 in doses 0.01 and 0.05 mg/kg,”actually doses ranged up to 1mg/ml

It is not clear why in fig 1a looking at intact joint diameter shown no change in swelling. It gives the impression that only the injected joint (1b) is swelling in response to injection rather than in response to inflammatory CFA model. I think the methods and nature of the model in the authors hands needs to be clearer. Was this used as a model of polyarthritis to model RA? Methods are insufficient to understand experimental design. Lewis strain is often used due to increased susceptibility for development of polyarthritis, but not in this study?. was this being used as a model of mono or polyarthritis? Usually adjuvant induced arthritis in the rat is a classic polyarthritis. How were time points chosen for this model? Was this done at peak disease severity? It would help to show longitudinal scoring results rather than at day 3. Generally this experiment could add to the paper, but unless a better set up of data from these animals is included, with decent supporting histology and validation of disease progression and therapeutic intervention this data could be left out and the study built around the MIA results. Particularly as the CFA is not the gold standard method for RA modelling, where CIA or TNFtg models might be more informative. Further limitations arise due to limited efficacy of NSAID interventions. Perhaps glucocorticoids would be better here, as no therapeutic effect is apparent. That being said, without optimising intervention timing around disease severity it is hard to properly judge data. Suggest removal

It would help to combine the key data from fig 1, 2 and 3 into 1 composite figure rather than presenting the variety of null data in its entirely. Much of this can feature in supplementary. One key concern is the lack of efficacy of diclofenac and ibuprofen in these experiments. Were doses sufficient. Where is supporting data on histology for their efficacy showing reduced synovitis or joint destruction? Again, further supporting data for these should be included (histology at least) to support observations. This would give some greater confidence in the results for this CFA model where many of the readouts are subjective and qualitative.

It appears the experimental intervention works for CFA, whilst NSAIDS offer no prevention from joint swelling, mobility or readouts of pain. This is a concern for this side of the data and needs to be addressed. Much of figs 1-3 is null data that could be in supplementary, whilst longitudinal data and histology are missing. Why day 3. Which joints were measured? Is this a measure of polyarthritis. Was joint inflammation at optimum to measure interventions. Statistical comparison within text

Greater explanation of what joint diameter ratio actually is and what it is measuring. Which joints/ joints

A large volume of text that looks like guidance for authors is pasted into the methods “Materials and Methods should be described with sufficient details to allow others to replicate 519 and build on published results. Please note that publication of your manuscript implicates that you 520 must make all materials, data, computer code, and protocols associated with the publication 521 available to readers. Please disclose at the submission stage any restrictions on the availability of 522 materials or information. New methods and protocols should be described in detail while 523 well-established methods can be briefly described and appropriately cited. 524

Research manuscripts reporting large datasets that are deposited in a publicly available 525 database should specify where the data have been deposited and provide the relevant accession 526 numbers. If the accession numbers have not yet been obtained at the time of submission, please state 527 that they will be provided during review. They must be provided prior to publication. 528

Interventionary studies involving animals or humans, and other studies require ethical 529 approval must list the authority that provided approval and the corresponding ethical approval 530 code.”

Methods for Hot plate test should be more descriptive, including reference to literature validating this approach.

  1. Conclusions appearing after methods seems unusual but may be standard for journal?

Fig 4. As with previous figures a lot of this data looks like dead weight, with null responses. Please move this data to supplementary and present the actual key data in main figs.

Fig 5. Key comparisons for interventions should be against saline. This comparison is the one that counts. Gig 5b appears to be missing error bars on several data points.

For MIA, the longitudinal data indicate that day 3 is optimal for analysis. Why not just present day 3 data for swelling and pain. Rest in supplementary. All pain recording data could be combined into one meaningful fig for day 3 MIA.

Fig 9 and 10. The histology scoring methods reported are insufficient to really assess their efficacy or reproduce. Histology data should be combined into 1 clear fig. images should support histograms, with clear areas highlighted. Difficult to really assess validity as they stand. Images appear as a separate fig. They are not of sufficient magnification or properly highlighted to correlate to histology scores and inform validity. Additional n numbers of histology should be considered for supplementary material to support the data. Better yet, quantitative micro-CT data of bones would really address joint destruction question.

Conclusions:

This interventions shows some merit but there is little evidence it outperforms standard control interventions. The discussion needs to be more measured. APHC3 may have merit, but it does not outperform on any level that is apparent in this study. It frequently, but not always matches the standard treatments. Its impact on inflammation is far from clear. The discussion and final conclusions should be written to reflect this more clearly and honestly.

Adjuvant-Induced Arthritis resolving is not gold standard for RA. CIA is and this should be clearly stated in discussion along with the limitations of the models utilised in this study

Author Response

In this study the authors explore the role of a TRPV1 inhibitor APHC3 in two models of arthritis (extrapolating to both RA and OA), exploring its impact on pain and inflammation. Aspects of the study are performed to a high standard such as measurements of pain. Other elements are limited or insufficient.  Particularly for the CFA model. This is used as an rheumatoid arthritis (RA) model and is not up to scratch. Further supporting data is needed to interpret their results in this model including histology, serum/synovial measurements of inflammation. It is not really clear whether it is being used as a model of polyarthritis monoarthritis, and its validity as a model of RA is not clear in this context. I would suggest removing this entirely from the manuscript and focussing on the more robustly assessed MIA model.

Complete Freund’s adjuvant (CFA)-induced rat monoarthritis model was used to evaluate analgesic/anti-inflammatory potential in the arthritis model. This is rather quick and robust model of arthritis pain generally corresponds to rheumatoid monoarthritis in human. It is an important part of the manuscript because it shows that APHC3 is capable to alleviate symptoms of arthritis independently of its genesis. We added a clear statement that this is the monoarthritis model in Abstract, Results, Discussion, and Methods. CFA-induced monoarthritis mimics well joint inflammation induced by the immune system, but not histological changes.  We considered it suitable for the evaluation of APHC3 potential as symptomatic treatment during RA.

In this manuscript, we evaluated the potential of APHC3 to alleviate the symptoms of arthritis.  APHC3 and NSAIDs caused a significant decrease in inflammation that evidently reduced the progression of the disease.  CFA – induced monoarthritis is an extremely studied and stable model that is used for validation of the efficacy of novel drugs (https://doi.org/10.1016/j.ejphar.2015.02.050) and verification of new methods for arthritis study (https://doi.org/10.1016/j.jneumeth.2017.04.011). It mimics the early stage of RA in human that often starts from acute inflammation of one joint In this model day 3 is the time point when acute inflammation begins to change to chronic and suits well for the assessment of inflammation and pain-related behavior (the standard pain-related behavior is described in https://doi.org/10.1016/j.jneumeth.2017.04.011 ,  Fig 7,8,10.

We added some information about CFA-induced monoarthritis in the manuscript -  lines 587-598

In addition, the authors need to combine their figs into fewer figs (11 is excessive for this paper), moving repetitive non-significant control data into supplementary would achieve this.

Figures were changed; a supplementary file with additional information was created.

 The histology in particularly needs to be presented more clearly and comprehensively to support the findings.

Figures were modified (Figure 9)

Lastly the discussion and final conclusions should reflect the merit of APHC3, but be more honest in the appraisal, as the evidence that it significantly outperforms standard NSAIDS here is almost entirely absent. There is some great data here, but it needs to be tidied up and made more concise.

We removed enthusiastic words from the conclusion.

Major

Introduction

Authors state “It is obvious that additional pharmaceutical targets should be explored for the OA and RA treatments, and one of the most prominent is TRPV1 channel”. I think that TRPV1 is an interesting target but in the field of rheumatology I wouldn’t say that it is the most prominent target. Perhaps reword for balance to potential or promising.

Changed to “promising”

In 2006 Barton et al explored the value of inhibition in a monoarthritis model, with some positive results, but this does not appear to have been extensively followed up in further models of arthritis

There were some studies with knockout mice confirming the findings of Barton, 2006. Additionally, the fact of clinical trials of TRPV1 antagonists on arthritis could be a proof of positive results obtained by some scientific groups that were not published.

2.1.1. Assessment of inflammation in vivo. Further clarity in reporting of the different doses is required. “APHC3 in doses 0.01 and 0.05 mg/kg,”actually doses ranged up to 1mg/ml

Corrected to 0.01, 0.05, 0.1 and 1 mg/kg

It is not clear why in fig 1a looking at intact joint diameter shown no change in swelling. It gives the impression that only the injected joint (1b) is swelling in response to injection rather than in response to the inflammatory CFA model. I think the methods and nature of the model in the authors hands needs to be clearer. Was this used as a model of polyarthritis to model RA? Methods are insufficient to understand experimental design. Lewis strain is often used due to increased susceptibility for development of polyarthritis, but not in this study?. was this being used as a model of mono or polyarthritis? Usually adjuvant induced arthritis in the rat is a classic polyarthritis. How were time points chosen for this model? Was this done at peak disease severity? It would help to show longitudinal scoring results rather than at day 3. Generally this experiment could add to the paper, but unless a better set up of data from these animals is included, with decent supporting histology and validation of disease progression and therapeutic intervention this data could be left out and the study built around the MIA results. Particularly as the CFA is not the gold standard method for RA modelling, where CIA or TNFtg models might be more informative.

This is the model of CFA-induced monoarthritis and the intact joint is not inflamed.  Intra-articular injection of CFA leads to infiltration of inflammatory cells and synovial hypertrophy and generally accepted as the RA model. This inflammatory model generates robust pain-like behavior therefore suitable for validation of anti-inflammatory and analgesic properties of novel drugs.  But it is important to note that it significantly differs from the histological point of view because common aspects of human RA such as bone erosion and cartilage serration are not frequently reported. CFA activates the innate immune system, but not adaptive, therefore, this model may not provide the optimal conditions for delineation of the mechanisms of RA development ( Bas 2016).  We added the explanation of this model and a discussion of its applicability in the text (lines 587-598).

Further limitations arise due to limited efficacy of NSAID interventions. Perhaps glucocorticoids would be better here, as no therapeutic effect is apparent. That being said, without optimising intervention timing around disease severity it is hard to properly judge data. Suggest removal

It would help to combine the key data from fig 1, 2 and 3 into 1 composite figure rather than presenting the variety of null data in its entirely. Much of this can feature in supplementary.

One key concern is the lack of efficacy of diclofenac and ibuprofen in these experiments. Were doses sufficient. Where is supporting data on histology for their efficacy showing reduced synovitis or joint destruction? Again, further supporting data for these should be included (histology at least) to support observations. This would give some greater confidence in the results for this CFA model where many of the readouts are subjective and qualitative. It appears the experimental intervention works for CFA, whilst NSAIDS offer no prevention from joint swelling, mobility or readouts of pain. This is a concern for this side of the data and needs to be addressed.

Data from figures 1 and 3, Figure 2 and null data moved to supplement.

The efficacy of ibuprofen was significant in the analysis of joint diameter ratio, heat hyperalgesia, and hind limb grip strength.  We consider this apparent therapeutic effect.  Doses were chosen according to previously reported results. We had no goal to reveal reasons for the inefficacy of diclofenac.  It should be effective according to the literature, but it was not and we just showed experimental data.  Negative results could help other investigators to interpret their data in this model.

Intra-articular injection of CFA leads to infiltration of inflammatory cells and synovial hypertrophy and generally accepted as the RA model. This inflammatory model generates robust pain-like behavior therefore suitable for validation of anti-inflammatory and analgesic properties of novel drugs.  But important to note that it significantly differs from the histological point of view because common aspects of human RA such as bone erosion and cartilage serration are not frequently reported. CFA activates the innate immune system but not adaptive therefore this model may not provide the optimal conditions for delineation of the mechanisms of RA development ( Bas 2016).  We added the explanation of this model and a discussion of its applicability in the text (lines 587-598).

Much of figs 1-3 is null data that could be in supplementary, whilst longitudinal data and histology are missing.

Data from figures 1 and 3, Figure 2 and null data moved to supplement. Histology data in this model are not informative due to the short period of the inflammatory process.

Why day 3.

CFA-induced monoarthritis is a well-established method. The pain and inflammation reach maximum at day 1 and 2 and at day 3 begin to reduce. Therefore, we started treatment at the peak of symptoms severity and analyzed the effects at the end of the maximum severity interval. Added to manuscript lines 587-598

 Which joints were measured?

 To make it more clear we modified section 4.6. Assessment of inflammation in vivo

Ankle joints diameters were measured in CFA-induced arthritis, and knee joints diameters – for MIA-induced arthritis. Joint diameters of both legs were measured using a digital caliper to evaluate swelling degree. Both absolute increase in joint diameter and the ratio between treated and intact joints (in percent of the intact joint) of the same animal were assessed. Joint diameter ratio was calculated according to the following equation: (diameter of injected joint/diameter of intact joint)*100.

Is this a measure of polyarthritis.

This is the monoarthritis model.  We added some additional information about CFA-induced monoarthritis in the manuscript - lines 587-598

Was joint inflammation at optimum to measure interventions.

CFA-induced monoarthritis is a well-established method. The pain and inflammation reach maximum at day 1 and 2 and at day 3 begin to reduce (please see the standard pain-related behavior assessment in https://doi.org/10.1016/j.jneumeth.2017.04.011 ,  Fig 7,8,10). Therefore we started treatment at the peak of symptoms severity and analyzed the effects at the end of the maximum severity interval.  The relevant information and references were added to discussion section

lines 587-598

Statistical comparison within text

Statistical comparison is shown in the figures. We tried to avoid duplication of data.

Greater explanation of what joint diameter ratio actually is and what it is measuring. Which joints/ joints

These data were stated in methods.

“Freund's Complete Adjuvant (CFA, 40 μl, Sigma-Aldrich) was injected intra-articularly to the right ankle joint with the left joint kept intact. The control group (CTRL) received intra-articular saline (40 μl) injection.”

we modified section  4.6. Assessment of inflammation in vivo

Ankle joints diameters were measured in CFA-induced arthritis, and knee joints diameters – for MIA-induced arthritis. Joint diameters of both legs were measured using a digital caliper to evaluate swelling degree. Both absolute increase in joint diameter and the ratio between treated and intact joints (in percent of the intact joint) of the same animal were assessed. Joint diameter ratio was calculated according to the following equation: (diameter of injected joint/diameter of intact joint)*100.

A large volume of text that looks like guidance for authors is pasted into the methods “Materials and Methods should be described with sufficient details to allow others to replicate 519 and build on published results. Please note that publication of your manuscript implicates that you 520 must make all materials, data, computer code, and protocols associated with the publication 521 available to readers. Please disclose at the submission stage any restrictions on the availability of 522 materials or information. New methods and protocols should be described in detail while 523 well-established methods can be briefly described and appropriately cited. 524

Research manuscripts reporting large datasets that are deposited in a publicly available 525 database should specify where the data have been deposited and provide the relevant accession 526 numbers. If the accession numbers have not yet been obtained at the time of submission, please state 527 that they will be provided during review. They must be provided prior to publication. 528 Interventionary studies involving animals or humans, and other studies require ethical 529 approval must list the authority that provided approval and the corresponding ethical approval 530 code.”

Text from the template was deleted from the manuscript.

Methods for Hot plate test should be more descriptive, including reference to literature validating this approach.

The fact of thermal hyperalgesia in models of arthritis is well-known

The method was described in more detail.

 “4.9.1. Hot plate test

Rats subjected to models of arthritis could develop hypersensitivity to heat [48,79]. Sensitivity to a thermal stimulus was tested using a Hot-Plate Analgesia Meter (Columbus Instruments, Columbus, OH, USA) set at 55 °C. Rats were placed individually on the preheated hot-plate surface and exposed to heat until a nociceptive reaction was registered. The test was discontinued if the withdrawal response was not observed for 30 sec. Pain threshold was detected as latency to hind paw withdrawal or licking.”

  1. Conclusions appearing after methods seems unusual but may be standard for journal?

We followed the manuscript template for Marine drugs.

Fig 4. As with previous figures a lot of this data looks like dead weight, with null responses. Please move this data to supplementary and present the actual key data in main figs.

The figure corrected. It is figure 2 in the revised manuscript.

Fig 5. Key comparisons for interventions should be against saline. This comparison is the one that counts. Gig 5b appears to be missing error bars on several data points.

We showed all statistically significant results, there is no significant difference from the saline group. There is a fine line between a variant of norm and pathology, and this is also significant if data corresponding to the variant of the norm.  We used a rigorous method for validation of statistical significance therefore comparison with two control groups was essential.  Figures were build-up by the statistical program, in this case, you can see the overlapping of maximum and interquartile rage.

For MIA, the longitudinal data indicate that day 3 is optimal for analysis. Why not just present day 3 data for swelling and pain. Rest in supplementary. All pain recording data could be combined into one meaningful fig for day 3 MIA.

Day 3 is an evaluation of single dose usage. It is not optimal for overall analysis of efficacy.  It is discussed in lines 652-671 of manuscript.

Fig 9 and 10. The histology scoring methods reported are insufficient to really assess their efficacy or reproduce. Histology data should be combined into 1 clear fig. images should support histograms, with clear areas highlighted. Difficult to really assess validity as they stand. Images appear as a separate fig. They are not of sufficient magnification or properly highlighted to correlate to histology scores and inform validity. Additional n numbers of histology should be considered for supplementary material to support the data. Better yet, quantitative micro-CT data of bones would really address joint destruction question.

 We added the reference about scoring in Methods.

 We added the suggested changes in histology figure 9.  Additional data on histology was added to the supplementary file.

Conclusions:

This interventions shows some merit but there is little evidence it outperforms standard control interventions. The discussion needs to be more measured. APHC3 may have merit, but it does not outperform on any level that is apparent in this study. It frequently, but not always matches the standard treatments. Its impact on inflammation is far from clear. The discussion and final conclusions should be written to reflect this more clearly and honestly.

 We removed enthusiastic words from the conclusion.  We claimed in conclusion " The efficacy of APHC3 was higher or equal to commonly used NSAIDs in standard tests on pain-related behavior." The merits and disadvantages of APHC3 are described in the discussion. Evidently, that meloxicam was better in the improvement of the histological score, while ibuprofen has close results in pain-behavior tests.

Adjuvant-Induced Arthritis resolving is not gold standard for RA. CIA is and this should be clearly stated in discussion along with the limitations of the models utilised in this study.

We added information about the correlation of CFA model of arthritis and RA.

Added to discussion lines 587-598

 “CFA activates the innate immune system but not adaptive therefore this model may not provide the optimal conditions for delineation of the mechanisms of RA development [52]. Collagen-induced arthritis is one of the most disease-related and widely used models of RA [52]. CFA-induced monoarthritis could be efficiently used for the analysis of novel anti-inflammatory and analgesic drugs suitable for arthritis symptomatic treatment [49]. Intra-articular injection of CFA leads to infiltration of inflammatory cells and synovial hypertrophy and is generally accepted as RA model. But important to note that it significantly differs from the histological point of view because common aspects of human RA such as bone erosion and cartilage serration are usually absent [52]. In the CFA-induced monoarthritis model, the pain and inflammation severity reaches the maximum at day 1 and 2 after CFA injection and at day 3 begin to reduce [51]. Therefore we started treatment at the peak of symptoms severity and analyzed the effects at the end of the maximum severity interval.”

Round 2

Reviewer 2 Report

Authors have extensivelly improved the manuscript. I still believe it is better to divide it in two, one for RA and other for OA, but the data is well described and discussion is supported by it.

In their anwers to my comment, they exert that they are focus on early arthirtis iefeects. This has to be written more clearly in the manuscript.

Author Response

Reviewer 2

Authors have extensivelly improved the manuscript. I still believe it is better to divide it in two, one for RA and other for OA, but the data is well described and discussion is supported by it.

In their anwers to my comment, they exert that they are focus on early arthirtis iefeects. This has to be written more clearly in the manuscript.

We added to line 373 “This simulates the early stage of RA in humans that often starts from acute inflammation of one joint.”

Reviewer 3 Report

Much of the non essential data have been relocated to the appendix and the message of the manuscript suitably tempered. However, the issue of excessive figures have not really been addressed. To me it is clear that figs 2-3 could be merged whilst 4-5 are measures of pain behaviour and could also be linked. figs 7-9 are all related histology and should be merged. The histology images are still of poor quality. Regions of interest should be orientated in similar positions between images and show comparable regions. Some currently show intra-articular space and meniscus whilst others show pannus bone interface. Close up of cartilage are required to clarify the damage being reported. Were cartilage stains available to support this.

Figs 8 a-c had missing error bars for standard deviation. Why are these absent? were n numbers complete here?

These formatting and presentation issues need to be addressed before publication

Author Response

Much of the non essential data have been relocated to the appendix and the message of the manuscript suitably tempered. However, the issue of excessive figures have not really been addressed. To me it is clear that figs 2-3 could be merged whilst 4-5 are measures of pain behaviour and could also be linked. figs 7-9 are all related histology and should be merged.

We would like to thank you for your careful attention to many details which greatly helped to improve this manuscript presentation and clarity. But we should note that there is no limitation on figures number in Marine drugs. Merging any of the proposed pairs makes them untidy and difficult to understand due to the excessive number of panels on one figure. The overall quality of these merged figures becomes much worse.  We believe that now the number, size, and composition of figures are comfortable for readers.

The histology images are still of poor quality. Regions of interest should be orientated in similar positions between images and show comparable regions. Some currently show intra-articular space and meniscus whilst others show pannus bone interface. Close up of cartilage are required to clarify the damage being reported. Were cartilage stains available to support this.

The histology illustrative material in the manuscript is presented strictly from an identical perspective. The hamstring is situated on the left side of the microphotograph, to the right of it - the synovial membrane and the nearest fragment(s) of the bones (femur or tibia). The synovial membrane of the knee joint is shown in the center of all the micrographs. The joint after decalcification was cut in the sagittal plane and both of its fragments (the medial one that was closer to the opposite joint and the lateral one) entered the subsequent processing with dehydration and embedding in paraffin. Thus, we obtained from one joint 2 “mirror” paraffin blocks, which were both subjected to microtomy, staining, and microscopy. This is the reason why the mirror image of the femur and tibia in different microphotographs could be observed.

We knowingly did not use any special processing of microphotographs to demonstrate clearly the morphofunctional features of knee joints tissues of experimental animals in the model of MIA-induced arthritis.

We did not use special methods of staining cartilage tissues.

Figs 8 a-c had missing error bars for standard deviation. Why are these absent? were n numbers complete here?

The number of samples is shown in figure legends. Figures were built up by the statistical program; in this case, you can see identical scoring of samples of some groups. If SD=0 there are no bars.

Round 3

Reviewer 3 Report

The authors have made no further requested changes to the quality or presentation of figures however I feel this is now an editorial decision rather than one as the reviewer